# Study on the Directional Solidification Process of an Aluminum Alloy Bar in Multishell Mold Being Gradually Immersed in Water

**DOI:** 10.3390/ma13092197

**Published:** 2020-05-11

**Authors:** Jiwu Wang, Lele Zheng, Jinwu Kang, Yongyi Hu

**Affiliations:** 1School of Mechanical, Electronic and Control Engineering, Beijing Jiaotong University, Beijing 100044, China; jwwang@bjtu.edu.cn (J.W.); 17121257@bjtu.edu.cn (L.Z.); 2School of Materials Science and Engineering, Tsinghua University, Beijing 100084, China; huyy@mail.tsinghua.edu.cn

**Keywords:** multishell mold, directional solidification, cooling, water level, insulation

## Abstract

A multishell mold structure and water-immersion cooling method (MSMWI) is proposed for the directional solidification of castings. A four-layer-shell sand mold was designed for a bar with diameter of 40 mm. As the aluminum melt was poured, the multishell mold was gradually immersed in water, and the water level drove the advancement of the solidification front from bottom to top. The multishell mold was helpful for the heat insulation of its upper part, and its bottom was chilled by the water. Therefore, directional solidification of the bar was vertically realized. The water-cooled solidification process of the bar was 5.8 times faster than that by air natural cooling (MSMNC), and the temperature gradient was increased by 78 times. The secondary dendrite arm spacing (SDAS) and eutectic silicon were significantly refined. Its tensile strength, elongation, and hardness were increased by 56%, 185%, and 62.6%, respectively.

## 1. Introduction

Casting cooling efficiency and quality control have always been the focus of casting research. Rapid solidification (RS) has been used to prepare metals, alloys, or metal glasses, which is a typical melt-quenching technique that can be classified into three categories: (i) spinning, (ii) droplet, and (iii) surface melting [1,2,3]. Metal dies were popularly used to replace traditional sand molds for higher cooling rates. Water-cooling and oil-heating channels are always arranged in the dies for high-pressure die casting [4,5]. During the making of billets, direct-chill (DC) casting was adopted to improve the cooling rate by bottom chill and water spray [6]. To further improve the cooling rate for large billets, intercooling was realized by inserting an in-mold cooler in replacement of the core rod into the melt sump [7]. Thus, the billet could be cooled from the center and surface positions. Zhang et al. used a phase-change medium to rapidly cool an A356 alloy to accelerate its cooling rate [8]. Nakayama et al. used a frozen mold to quicken the cooling of castings [9]. Directional solidification has been widely used in the investment-casting process for the production of turbine blades [10,11], in which the bottom of castings is cooled by a water brass jacket, the lower part in the cold chamber and the upper part kept in hot chamber. In recent years, the liquid-metal-cooling (LMC) method was proposed to replace the cold chamber for a greater temperature gradient [12,13]. Grassi et al. proposed an ablation method in which water washed away the sand mold from one end to another to improve cooling efficiency [14]. The effect of water cooling on the microstructure and mechanical properties of castings was investigated [15,16]. Kang et al. proposed the post solidification-intensive-riser-cooling (PSIRC) method for heavy castings, and achieved their fast and even cooling [17]. The application of the additive-manufacturing method in castings provides more freedom for the design of castings, sand molds, and cores, for example, in the design of the skeleton structure of parts and lattice structures, core consolidation, and gating-system optimization, including pouring basins, sprues, and ingates [18,19,20,21]. Shangguan et al. proposed a new mold design, the hollow sand mold, which had great potential for the cooling control of the solidification process of castings [22]. Deng et al. proposed a hollow-shell design for heat insulation [23]. Kang et al. carried out a numerical simulation of the hollowed-out sand-mold structure and showed that the structure could significantly improve cooling speed, and reduce the stress and deformation of castings [24].

In this study, a multishell mold structure and water-immersion cooling method are proposed that were used to realize directional solidification from the bottom to the top of a bar. Its temperature gradient, microstructure, and properties were investigated.

## 2. Experiment Procedures

To realize the directional solidification of a casting, a multishell mold structure and water-immersion cooling method (MSMWI) is proposed, as shown in Figure 1a. After pouring, the mold was cooled in a water tank with the gradual rise of water level. Water flowed into the shell spacings and surrounded the innermost shell, which provided good cooling capability and drove the solidification front upward. Meanwhile, the multishell provided insulation function for the upper part of the casting.

### 2.1. Design and Manufacture of Multishell Sand Mold

To investigate the directional solidification process of the MSMWI method, a bar with diameter of 40 mm and height of 150 mm was selected for investigation. A 50 mm high riser was added. For the heat dissipation of the multilayer-shell sand mold during the casting process, the air gaps served as insulation layers, which reduces the heat transfer from casting to mold and also that from mold to the environment. With the increase of the number of shell layers, the thermal resistance of the mold increased, which helped to hinder heat transfer inside the mold. As the number of shells increased to a certain number, the insulation effect was slowly improved [25]. Considering both the thermal-insulation effect and the convenience of the experiments, a four-layer-shell mold design (with three layers of gaps) was adopted, as shown in Figure 1b. There were four shell layers in total of 5 mm thick each, and three layers of cavities of 10 mm width each. The multishell mold consisted of body and cover. The multishell mold could be poured into the mold from the top with the cover removed, and the cover could be placed back as the pouring finished. These multishell sand molds were produced by using an ExOne S-Max printer (Gersthofen, Germany) based on the binder-jetting 3D-printing (3DP) method. Quartz sand with a particle diameter range of 75–150 μm was used with 1.8% addition of furan resin binder.

### 2.2. Experiment

The A356 aluminum alloy (composition listed in Table 1) was melted and kept at 720 °C. The melt was poured into two identical multishell molds from the top, and then they were covered with the multishell insulation cover. After pouring, one mold was cooled by air natural cooling (MSMNC), and the other was moved into a water tank (MSMWI). Four thermocouples (P1–P4) were set to monitor cooling at different heights of each bar, as shown in Figure 2a,b.

According to the values of P1–P4 of MSMWI, water level rose to push the solidification front upward, and water level was always 5 mm lower than the solidification front. A water-level control platform for directional solidification was designed and set up as shown in Figure 3a. An ultrasonic level sensor was used to monitor the water level. The four thermocouples and water-level sensor were connected to a Siemens s7200 smart PLC (Berlin, Germany) for process control. The PLC controlled the work of a pump by comparing the current solidification front deduced from P1–P4 and the current water level. Water-level history is shown in Figure 3b. The temperature of the multishell mold was measured by using a Flir T250 infrared thermal imaging camera (FLIR Systems, Orlando, FL, USA). The cast bars were machined into samples for microstructure observation and hardness test, and the tensile bars for tensile tests according to Chinese National Standard GB/T228 [26]. MTS E45.105 (MTS Systems Corporation, Eden Prairie, MN, USA) was used in the tensile tests. For casting-hardness (HBW) measurement, we used the HBRV-187.5 hardness tester (Lunjie, Shanghai, China). The aluminum samples were corroded with 0.5% HF for 15 s, and the microstructure of the sample was observed with Zeiss FE-SEM (Oberkochen, Germany).

## 3. Results and Discussion

### 3.1. Cooling and Solidification

The cooling curves are shown in Figure 4a. It took the casting 27.5 min to be cooled down to 400 °C by MSMNC, while only 4.75 min by MSMWI; cooling efficiency was improved by 5.8 times. Figure 4b gives the temperature-gradient variation of the two castings in the height direction. The casting had a great temperature gradient up to 2.86 °C/mm with MSMWI, while this was less than 0.1 °C/mm with MSMNC. The large temperature gradient helped the MSMWI casting achieve sequential solidification from bottom to top.

The temperature profiles of the multishell mold under two cooling conditions and the thermal histories of typical locations of the molds are shown in Figure 5a,b. There were two different states of the shell mold under water cooling; below water level, the shell mold was at room temperature, while above water level, it was around 100 °C. Water cooling accelerated the solidification and cooling of the casting at the bottom, which led to the directional solidification of the bar from bottom to top. 

Under MSMNC, the highest temperature of the mold body was located at the area corresponding to the reinforcement ribs, 20 °C higher than the surface corresponding to the hollow spacing, which meant the multishell structure had an insulation effect.

Under MSMWI, the whole multishell mold was quickly heated up within 5 min. The reason is that water vapor generated by the boiling water by the immersed part of the mold soon filled the whole spacing between shells, including the mold cover, and then it heated the whole multishell mold. This was helpful for postponing the solidification of the upper part. The evaporation of water and its heating effect need further investigation.

### 3.2. Microstructures

The microstructures of the castings are shown in Figure 6. In the MSMNC casting, α-Al grains were coarse with rough boundaries, and the eutectic silicon was large, with irregular and sharp contour, around 10–50 μm long. However, in the MSMWI casting, the alpha dendrites showed round and typical secondary arms, the eutectic silicon phase was fine particles, around 2–8 μm in diameter, around a fiftieth of that in the MSMNC casting. 

The measured secondary dendrite arm spacing (SDAS) of castings at points O1‒O4 is shown in Figure 7a and listed in Table 2. The SDAS of the MSMWI casting was in the range of 47.4–65.7 μm, only 60–70% of the range of 72.1–98.6 μm of the MSMNC casting. The fine matrix grains and eutectic silicon particles were helpful for the improvement of mechanical properties, especially the elongation rate. A356 is a hypoeutectic alloy. During the solidification process, the primary α-Al phase first appeared, and then the eutectic phase precipitated according to the aluminum–silicon eutectic reaction at the eutectic temperature of 577 °C. Compared with MSMNC, the larger temperature gradient and higher cooling rate in MSMWI increased the undercooling degree during casting solidification; thus, the nucleation and grain-growth rates in the aluminum alloy melt were greatly increased, which resulted in fine matrix grains and the eutectic silicon phase.

### 3.3. Mechanical Properties

The stress–strain curves of MSMNC and MSMWI are shown in Figure 7b, and tensile-strength, yield-strength, elongation, and hardness data are listed in Table 2. Compared to MSMNC, the mechanical properties of MSMWI were obviously improved, tensile strength increased by 56%, yield strength by 20%, elongation by 185%, and hardness by 62.6%. For thick castings, the enhancement of mechanical properties is more significant.

The casting using the MSMWI directional solidification process had a higher average temperature gradient, which played a key role in the growth of the casting microstructure. MSMWI significantly refined the grains and reduced the SDAS of the structure, thereby improving the mechanical properties of the casting.

## 4. Conclusions

The directional solidification process of the bar casting was realized by gradually immersing the poured multishell sand mold into a water tank (MSMWI). Compared with natural cooling (MSMNC), it greatly improved cooling efficiency by 5.8 times, and temperature gradient by 78 times. The SDAS of alpha grains and the eutectic silicon phase were significantly refined, and the mechanical properties were greatly improved, especially elongation, by 185%. The proposed MSMWI method is helpful for the directional solidification of alloys.

## Figures and Tables

**Figure 1 materials-13-02197-f001:**
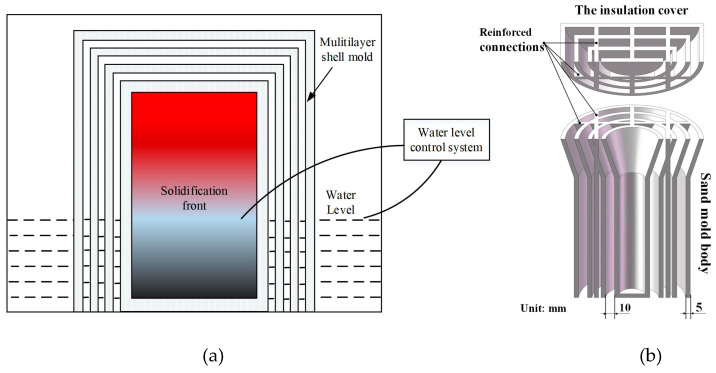
(**a**) Multishell-mold structure and water-immersion cooling method (MSMWI); (**b**) four-layer-shell sand mold.

**Figure 2 materials-13-02197-f002:**
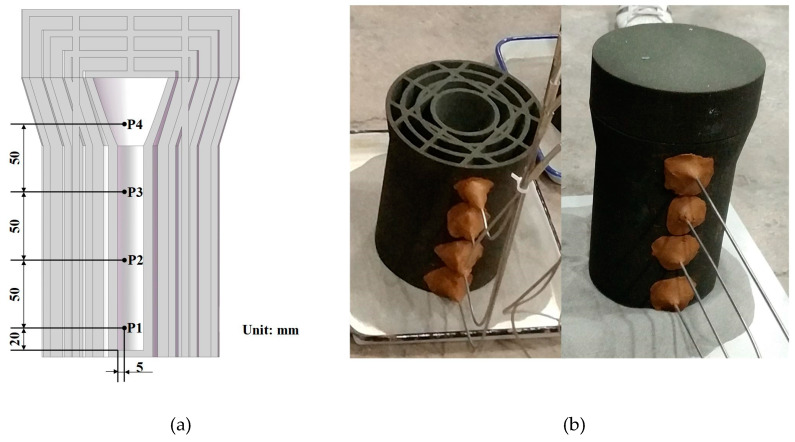
(**a**) Thermocouples (P1–P4); (**b**) 3D-printed sand mold and thermocouples.

**Figure 3 materials-13-02197-f003:**
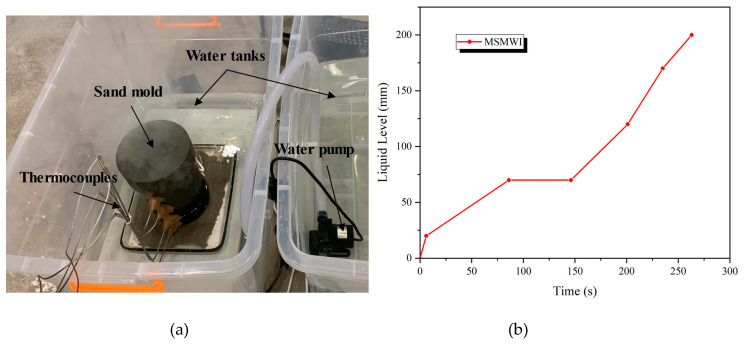
(**a**) Poured multishell sand mold cooled in water tank; (**b**) water-level history for MSMWI.

**Figure 4 materials-13-02197-f004:**
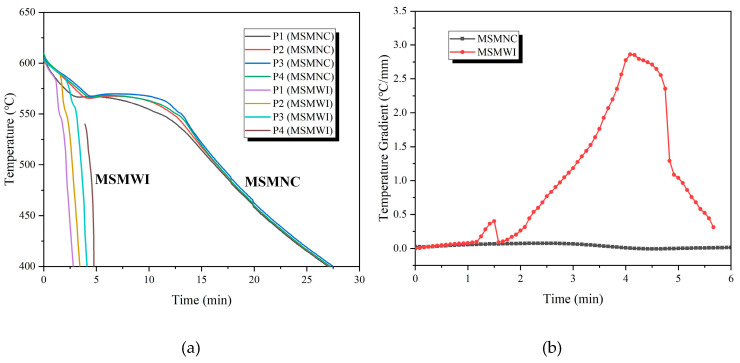
(**a**) Casting cooling curves; (**b**) temperature gradient of castings vs. time.

**Figure 5 materials-13-02197-f005:**
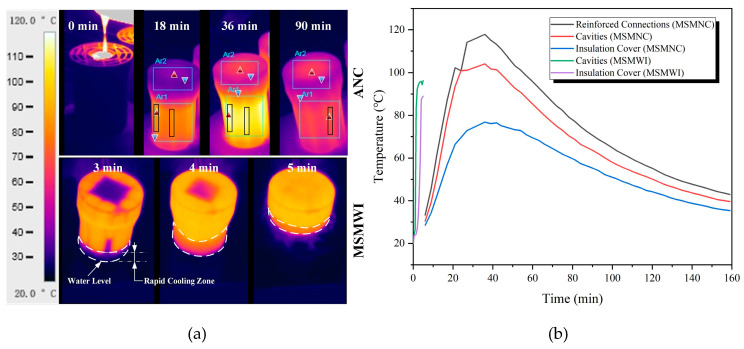
(**a**) Mold thermal images; (**b**) temperature curves of multishell molds by air natural cooling (MSMNC) and MSMWI.

**Figure 6 materials-13-02197-f006:**
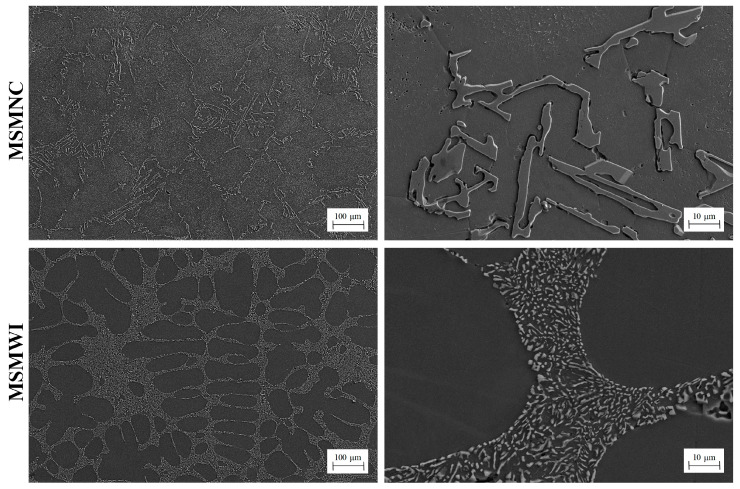
Microstructure of MSMNC and MSMWI.

**Figure 7 materials-13-02197-f007:**
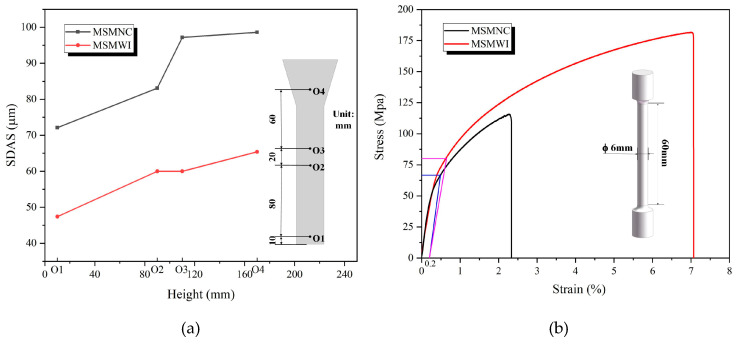
(**a**) Secondary-dendrite-arm-spacing (SDAS) variation; (**b**) stress–strain curves.

**Table 1 materials-13-02197-t001:** Chemical composition of A356.

Element (wt.%)							
Si	Mg	Fe	Cu	Zn	Mn	Ti	Al
6.5–7.5	0.25–0.45	0.12	0.05	0.05	0.05	0.08–0.20	Remaining

**Table 2 materials-13-02197-t002:** SDAS, mechanical properties, and hardness.

Items	SDAS (μm)				UTS (MPa)	0.2% PS (MPa)	Elongation (%)	Hardness (HBW)
	O1	O2	O3	O4				
Casting 1	72.1	83.1	97.2	98.6	116	67	2.6	24.3
Casting 2	47.4	60	60	65.4	181	81	7.4	39.5

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
