# Peer review of "Study on the Directional Solidification Process of an Aluminum Alloy Bar in Multishell Mold Being Gradually Immersed in Water"

_materials, 2020, doi:10.3390/ma13092197_

Round 1

Reviewer 1 Report

The experimental results are interesting for understanding effect of directional solidification on microstructure and mechanical properties of A356 castings.

Pore formation is mainly concerned and greatly affect the tensile properties of A356 alloy castings. If casting can be designed with proper control in cooling (growth rate x temperature gradient at the solidification front) by directional solidification,  the pore size and SDAS can be reduced to therefore enhancing tensile properties.

Fig. 1 is complicated and  need to remove Fig. 1a, 1b and 1e for improving its clarity.

Fig. 2 and Fig 3 must improve the clarity for reading.

Introduction:

Author must explain why three layers of shell core sand molds were used for study instead of two layers or one layer shell core sand molds.

Experimental procedure:

1. The authors did not adopt degassing treatment for removing hydrogen in this study, explain why?

For evaluating tensile properties of aluminum alloy castings, degassing treatment is commonly used for enhancing tensile properties.

2. Hydrogen content should be measured to point out the quality of melt before pouring. 

Results and discussion:

Authors must extend their discussion about effects of R (growth rate) and G (temperature gradient) on the SDAS(second dendrite arm spacing) and eutectic silicon size. Encourage to search more data from literatures and make a comparison of R and G on the microstructure of A356 alloy castings and pore formation (fractions).

Reviewer 2 Report

40mm and all next units - 40 mm,

20℃ (degrees) and 60% (percentages) are correct,

Fig. 2 - font too small, poor quality, a-d symbols hard to find in Fig.,

Fig. 3 - font too small,

Mpa – MPa,

No information was found as determined by the temperature gradient (in which place - probably not represented by a linear function for the entire chamber of the mold). Why was the shape of the casting chamber used? Why three water coats? This requires clarification at work. The methodology of the experiment requires a more detailed description.

Table 1 – font.

Reviewer 3 Report

This is a simple piece of research and is economically but adequately reported. It reports what is perhaps best described as a piece of technology which can benefit from modern 3-D printing of sand molds. There is minimal science reported here, and no optimisation of the technology (no study of different rates of immersion, different temperatures, different layers of shell etc. 

One detail which would benefit from clarification is

page 1 line 29 'water brass jacket'. What does this mean? Does it refer to 'a water-cooled copper jacket'.  Perhaps the water-cooled jacket may be made of brass. 

The 'Conflict of Interest' was not signed. 

Round 2

Reviewer 1 Report

2.2 Experiment

Line 94

The aluminum samples were corroded with 0.5% HF for 15 s ---.

=> etched to replace corroded